# Application of Nanotechnology in COVID-19 Infection: Findings and Limitations

**Ibrahim A. Shehu** [1,*]🆔**, Muhammad K. Musa** [2]🆔**, Aparna Datta** [3]🆔 **and Amita Verma** [4]

1   Department of Pharmaceutical Services, Murtala Muhammad Specialist Hospital, Hospital Management Board, Kano 700224, Nigeria
2   Department of Pharmaceutical Sciences, Ahmadu Bello University, Zaria 810107, Nigeria
3   Department of Pharmaceutical Technology, School of Health Science, NSHM Knowledge Campus, Kolkata 700053, India
4   Bioorganic and Medicinal Chemistry Research Laboratory, Department of Pharmaceutical Sciences, Sam Higginbottom University of Agriculture, Technology and Sciences, Prayagraj 211007, India
*   Correspondence: ibrahimaminushehu@gmail.com

**Abstract:** There is an urgent need to address the global mortality of the COVID-19 pandemic, as it reached 6.3 million as of July 2022. As such, the experts recommended the mass diagnosis of SARS-CoV-2 infection at an early stage using nanotechnology-based sensitive diagnostic approaches. The development of nanobiosensors for Point-of-Care (POC) sampling of COVID-19 could ensure mass detection without the need for sophisticated laboratories or expert personnel. The use of Artificial Intelligence (AI) techniques for POC detection was also proposed. In addition, the utilization of various antiviral nanomaterials such as Silver Nanoparticles (AgNPs) for the development of masks for personal protection mitigates viral transmission. Nowadays, nano-assisted vaccines have been approved for emergency use, but their safety and effectiveness in the mutant strain of the SARS-CoV-2 virus remain challenging. Methodology: Updated literature was sourced from various research indexing databases such as PubMed, SCOPUS, Science Direct, Research Gate and Google Scholars. Result: We presented the concept of novel nanotechnology researched discovery, including nano-devices, electrochemical biosensing, nano-assisted vaccine, and nanomedicines, for use in recent times, which could be a formidable step for future management of COVID-19.

**Keywords:** COVID-19; nanomaterials; nanobiosensors; sensing techniques; (AI) techniques; nano-assisted vaccines; and nanomedicines

## 1. Introduction

Coronaviruses are the largest family of single-stranded RNA-enveloped viruses [1]. The SARS-CoV-related family is the most severe type of coronavirus that causes chronic lower respiratory tract infections and acute respiratory distress syndrome (ARDS) [2,3]. The novel SARS-CoV (COVID-19) outbreak at the Wuhan market of China, which was earlier diagnosed as pneumonia of unknown aetiology before the International Committee on Virus Taxonomy (ICVT), proved its 80% genetic resemblance to SARS-CoV version 2002, and therefore, it was named SARS-CoV-2. The exact transmission mode of SARS-CoV-2 remains unclear; although, human-to-human, airborne and faecal-oral transmissions have been reported [4,5]. The World Health Organization (WHO) declared SARS-CoV-2 a pandemic disease and public health emergency because of its high contagiousness and mortality rates [6–9]. As of 7th September 2022, 228 countries and territories were affected, and over 600 million people were affected, with global mortality of over 6.5 million [10–12].

The current literature indicates that the employment of nanoparticles and nanomaterials has been useful in the development of rapid and accurate diagnostic devices, novel vaccines, and nanoscale fibres that enhance the protective affinity of personal protective

equipment (PPE) [13–15]. Currently, nanotechnology tools have revolutionized the strategies for designing SARS-CoV-2 vaccines [16]. For instance, the Moderna, Pfizer/BioNTech vaccines were developed by encapsulation of messenger Ribonucleic Acid (mRNA) in Lipid Nanoparticles (LNPs), enabling them to achieve desirable delivery efficiency, safety and effectiveness [17]. All the live attenuated and inactivated viral vaccines are regarded as nanoparticles themselves. In addition, they can be employed as an excellent delivery vehicle for nucleic acid and other viral fragments [18].

This review aims to emphasize the collective roles and challenges of employing nanotechnology to build quick and highly precise testing equipment and electrochemical biosensing supported by AI beneficial for early-stage diagnosis and treatment of COVID-19.

## 2. Possible Nano-Assisted Druggable Targets of COVID-19 Life Cycle

The table below illustrates some important drug-targeted sites in the SARS-CoV-2 life cycle, including the cellular entry mechanism, involving angiotensin-converting enzymes (ACE2), transmembrane serine protease 2 (TMPRSS2), RNA transcription, replication mediated by helicase and RNA-dependent RNA-polymerase (RdRp). The table further describes the translation and proteolytic processes of viral proteins mediated by chymotrypsin-like and papain-like proteases in addition to virion assembling and exocytosis [19]. These forgettable proteins are essential for curing COVID-19 with existing drugs that have therapeutic potency against these specific proteins and enzymes [20]. Drugs capped with silver Nanoparticles (AgNPs) were found to have the potential to block the cellular entry of SARS-CoV-2 through ACE2 attachment [21]. The development of recombinant human-ACE2 (rhACE2) for the treatment of acute lung injury has now shown to be a powerful viral entry inhibitor [22]. The life cycle and druggable targets of SARS-CoV-2 are schematically presented in Figure 1, while the mechanisms of targeting are summarized in Table 1.

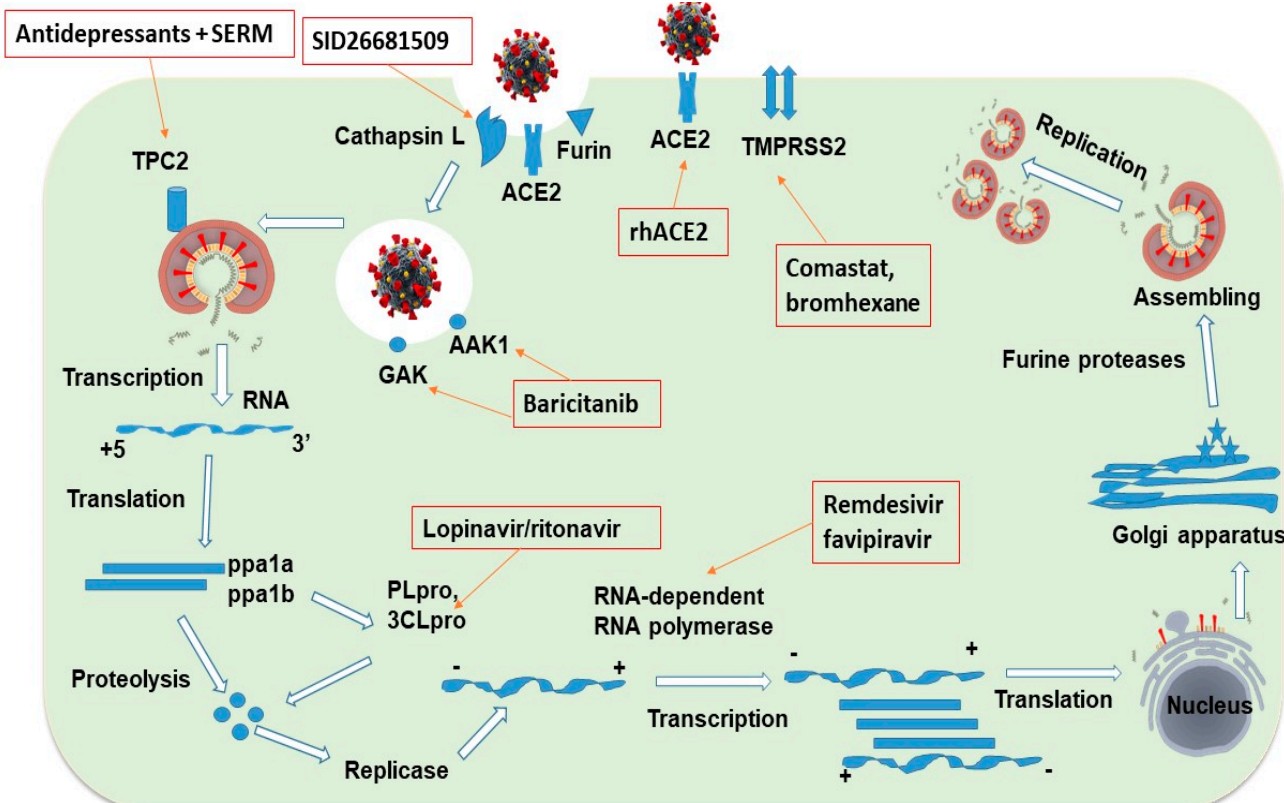

**Figure 1.** Schematic presentation and druggable target of SARS-CoV-2 (the figure is drawn by the author using Microsoft PowerPoint).

**Table 1.** Possible nano-assisted druggable targets of COVID-19.

| Nano-Assisted Drugs | Mechanism |
| --- | --- |
| 1. ACE 2 receptors | It has been confirmed that SARS-CoV-2 enters cells via the ACE2 receptor. This implies that the blockage or modulation of ACE2 could be a potential means to halt the cellular entry of SARS-CoV-2 and prevent subsequent infection. The mechanism of blockage involves either inhibiting the virus ACE2-RBD interaction, directly inhibiting ACE2-receptors, or using recombinant human ACE2 (rhACE2); APN01 to disrupt virus endocytosis [21]. |
| 2. TMPRSS2 | Human TMPRSS2 more powerfully activates the COVID-19 spike protein and facilitates its fusion across the host cell membrane via ACE2 than endosomal cathepsin and other protease enzymes. As a result, targeting TMPRSS2 may be critical for developing anti-SARS-CoV-2 molecules. Some potent TMPRSS2 inhibitors, such as camostate and bromhexine, have been shown to be repurposed against COVID-19. Stopsack K investigated the anti-TMPRSS2 activity of androgens and other corticosteroids and demonstrated their potential for repurposing in the treatment of COVID-19. Furthermore, determining the crystal structure of the TMPRSS2 protein could aid in the development of COVID-19 [23]. |
| 3. Furin | The SARS-CoV-2 S-protein possessed a Furin-like cleavage site (FCS), which is responsible for the S-structural protein's form, electrostatic interaction, and furin binding affinity. The study in Zhejiang province examined the effect of FCS mutation (F1-2) Hasan A and demonstrated the impact of furin-ACE2 enzyme interaction and its pathogenic role in COVID-19 infection, determining its relative proclivity to influence the structure of S-protein and the interaction between FCS and Furin [24]. |
| 4. Capthepsin L | The activation of S-protein by lysosomal cathepsin facilitates COVID-19 cellular fusion and endocytosis. Targeting the cathepsin L enzyme could be a therapeutic option for developing novel anti-coronavirus moiety. The Hoffmann M study demonstrated the effectiveness of cathepsin L inhibitor (SID26681509) in vitro and concluded that the compound could selectively prevent pseudovirus cellular entry by 76 % [25]. It may help to prevent the progression of pulmonary fibrosis. Furthermore, combining cathepsin L with TMPRSS2 inhibition to target COVID-19 could provide an effective cure for coronavirus infection. However, the lack of a crystal structure for cathepsin L could aid in the development of specific antibodies against the cathepsin L enzyme and thus the COVID-19 infection [26]. |
| 5. Two-Pore Channel (TPC2) | This is a voltage-gated channel essential for sodium and calcium exchange across the cell membrane, and it was reported to play a vital role in cellular trafficking of the Ebola virus [27]. The combination of antidepressants (such as pimozide and fluphenazine) and selective estrogen receptor modulators SERM (such as raloxifene, tamoxifen and clomiphene) is reported to have shown a potential blockage effect on TPC2 [28], and hence, it could be regarded as a possible target to curb COVID-19 infection. |
| 6. AAK1 and GKA | AAK1 and GKA are serine-threonine kinase enzymes responsible for virus intracellular endocytosis and genomic release in hepatitis c, Ebola, Dengue and prospectively COVID-19 viruses [29]. Baricitanib, an AAK1 and GKA inhibitor, was suggested for the treatment of COVID-19 by interfering with the virus's cellular entrance mechanism, although the proposal lacked clinical validation. However, the clinical trial conducted in Europe for the use of Baricitanib in COVID-19 patients has shown a negative prognosis [30]. Structural-based studies of AAK1 and GKA revealed the possibility of developing or modifying drug molecules effective for the treatment of COVID-19 infection [31]. |
| 7. Nonstructural protein: | Coronavirus possessed 16 various nsp rendering functions, although the specific functions elicited by nsp remain unclear. The crystal structure of nsp has been established, aiding the development of novel molecules in blocking its action [32,33]. |

**Table 1.** *Cont.*

| Nano-Assisted Drugs | Mechanism |
|---|---|
| 8. Proteases 3CLpro and PLpro | The COVID-19 main proteases 3CLpro and PLpro have a vital role in its replication processes. Inhibition of 3CLpro and PLpro with peptides and peptidomimetics would be a hot target for COVID-19 prevention [34]. Previously, the FDA approved two potent inhibitors of 3CLpro (lopinavir and ritonavir) as repurposed drugs for use against COVID-19 [35]. |
| 9. RNA-Dependent RNA Polymerase (RdRp) | RdRp plays a crucial role in facilitating the process of genome replication and transcription in the COVID-19 life cycle [36]. The development of the RdRp structure in complex with its cofactors (nsp7 and nsp8) is encouraging. Therefore, interfering with the function of these enzymes would help the drug design for the treatment of COVID-19 infection [37]. |

## 3. Nanotechnology for Efficient COVID-19 Diagnostics

Clinical applications of nanotechnology have revolutionized the pharmaceutical and biotech industries, especially in the past two decades. It permits the formulation of nano vaccines, nanomedicines, nano-vectors, and sensing devices for the diagnosis, prevention, and treatment of many diseases [38]. However, the impact of nanobiosensors in the biomedical aspect is very significant. It is widely employed to detect viruses and bacteria present at minimal concentrations, which may aid in the rapid detection of the SARS-CoV-2 pandemic in the sample collected from asymptomatic patients. The detection can easily be achieved since the nano biosensors coupled with an isothermal amplification assay can confirm the presence of COVID-19 in less than 40 min [39,40].

The surface engineering of single and hybridized nanoparticles such as photonic crystals, metallic nanoparticles, graphene, carbon nanotubes (CNTs), and photonic crystals have been efficiently used for viral detection and, thus, could be applied to COVID–19 [41,42]. For instance, CNT-based biosensors were developed to detect the nucleocapsid protein of COVID-19 with high sensitivity in the sample [43]. However, photonic crystals functionalized with amino and carboxyl functional groups can detect SARS-COV-2 spike protein [44]. An oligonucleotide of SARS-CoV viruses is detectable by a "surface plasmon resonance-based biosensor", whereas the N-protein is specifically detectable using "single-stranded DNA aptamers" [45,46].

Gold nanoparticles (Au-NP) have a wide range of biomedical applications; they interact with various forms of viruses because of their unique electric, photolytic and catalytic properties. Consequently, Au-NP has been employed for viral detection assays. Recently, V.X. Ting and his team developed an AuNP-based biosensor that detects the presence of COVID-19 in the sample within 5 min [47,48]. Layqah et al. proved that Au-NP-based biosensors possessed high precision in detecting COVID-19 in a few minutes [49]. It was evident that graphene oxide nanoparticles (GONPs) and silver-graphene nanoparticles (Ag-GNPs) have a high affinity for inhibiting the interaction between the S-protein and host cells [50,51]. Additionally, zinc nanoparticles (Zn-NPs) claimed to have interfered with the viral replication process while inducing interferon production. Therefore, Zn-NPs could be beneficial in the combat against COVID-19 [52,53].

### 3.1. Nucleic Acid-Based Sensing Mechanism

Nucleic acid-based sensing is the conventional means of clinical detection of RNA viruses [54]. The numerous functional and structural proteins possessed by SARS-CoV-2 could be essential targets for inhibition, sensing, and diagnosis. Likewise, the single-stranded RNA genome is another key target for viral testing [55–57], which could equally help in testing for COVID-19. These sensing systems have shown significant effects in detecting the COVID-19-RNA at a minimal concentration within a short period, and this could be advantageous over Real-Time Polymerase Chain Reaction (qRT-PCR) testing because nucleic acid-based sensing is handier and more economical than qRT-PCR. For instance, Recombinant Polymerase Amplification (RPA) techniques can now be applied to achieve

rapid detection of COVID-19 due to it is specificity and reliability in the amplification of viral recombinase and polymerase enzymes at room temperature [58–60].

The detection of SARS-CoV-2 virus RNA is the most important aspect of SARS-CoV-2 molecular diagnostics. However, the detection capacity was limited by several challenges, including the collection of dynamic and lower viral load samples during infection, sampling and handling issues, and the impact of hormonal responses to COVID-19 infection [61]. Molecular detection of RNA includes antigen-based and antibody-based sensing systems, as discussed below.

### 3.1.1. Antigen-Based Sensing

Detection of viral antigens is another method of diagnosing viral infections, including COVID-19. Although, the method is difficult to achieve due to the possible viral load fluctuation in the patient and high antibody responses [62]. Bin Ju et al. examined 15 and 206 samples collected from COVID-19 patients, respectively, and found a high amount of specific antibodies against SARS-CoV-2 RBD [63]. According to Zhao et al., an indirect ELISA kit based on recombinant nucleoproteins was tested and found to be 99% sensitive to SARS-CoV-2 antibodies (IgM and IgG). Currently, the FDA has approved more than 20 kits for use in an emergency to detect COVID-19 virus proteins [64]. The rapid antigen detection kit has been validated by the WHO and approved by the FDA for use in the diagnosis of COVID-19 because of its ease of handling, cheapness and speedy detection capability of less than 1 h. It was demonstrated to have given a limited sensitivity and high false negative result. Therefore, the rapid antigen detection assay for COVID-19-positive samples requires further validation with PCR testing before isolation and treatment [65]. The schematic presentation of the molecular diagnosis of COVID-19 is highlighted in Figure 2.

The sophisticated biosensors are designed to detect COVID-19-specific antigens present at a minute concentration in the suspected sample. However, some nanomaterials are found to have shown ideal properties for detecting molecules; these materials have been used to design sensitive biosensors for the detection of specified antigens [66]. Nowadays, graphene nanomaterial has been widely employed to develop nano biosensors against COVID-19 because of its unique properties of being conductive and 2-dimensional in nature (2D) [67]. Recently, graphene Leaders Canada Inc. (GLC), in partnership with GLC-Medical (GLCM) Inc., created the first graphene-based testing kit for COVID-19. The kit provides rapid and accurate testing capacity, specifically for SARS-CoV-2 antigen, with no overlapping of the result with other n-CoV-related viruses and economic benefits to the patients [68]. Nevertheless, the Health Canada and United States Food and Drug Administration (US-FDA) approved the first sliver test for point-of-care diagnosis of COVID-19. However, it would clinically replace both nucleic acid and serological tests, including the nasopharyngeal swab test for COVID-19 detection [69].

### Reverse Transcription-Polymerase Chain Reaction (RT-PCR) Assays

The protein targeted for the assay determines the analytical specificity and sensitivity of RT-PCR. The relative specificity of each SARS-CoV-2 specific protein varies, making COVID-19 identification easier. For example, the E, N, S, and RdRp genes distinguish SARS-CoV-2 from other SARS-CoV viruses. German researchers created the first and most sensitive RT-PCR assays targeting SARS-E, CoV-2's N, and RdRP genes [70]. In early 2020, the WHO approved seven RT-PCR kits for detecting SARS-CoV-2 in analytical samples, and the FDA and EUA later approved approximately 25 products. The RT-PCR assay that was later developed to target the N protein was 10 times more sensitive than the assay that targeted the Orf1b gene. As a result, it was commercialized by the Foundation for the Innovation of Research Diagnostics (FIND) for the rapid detection of SARS-CoV-2. However, the detection capacity of RT-PCR is limited by the high rate of false negative results caused by sampling and handling errors, as well as the lack of RT-PCR kits in rural areas and its inability to detect asymptomatic patients [71,72].

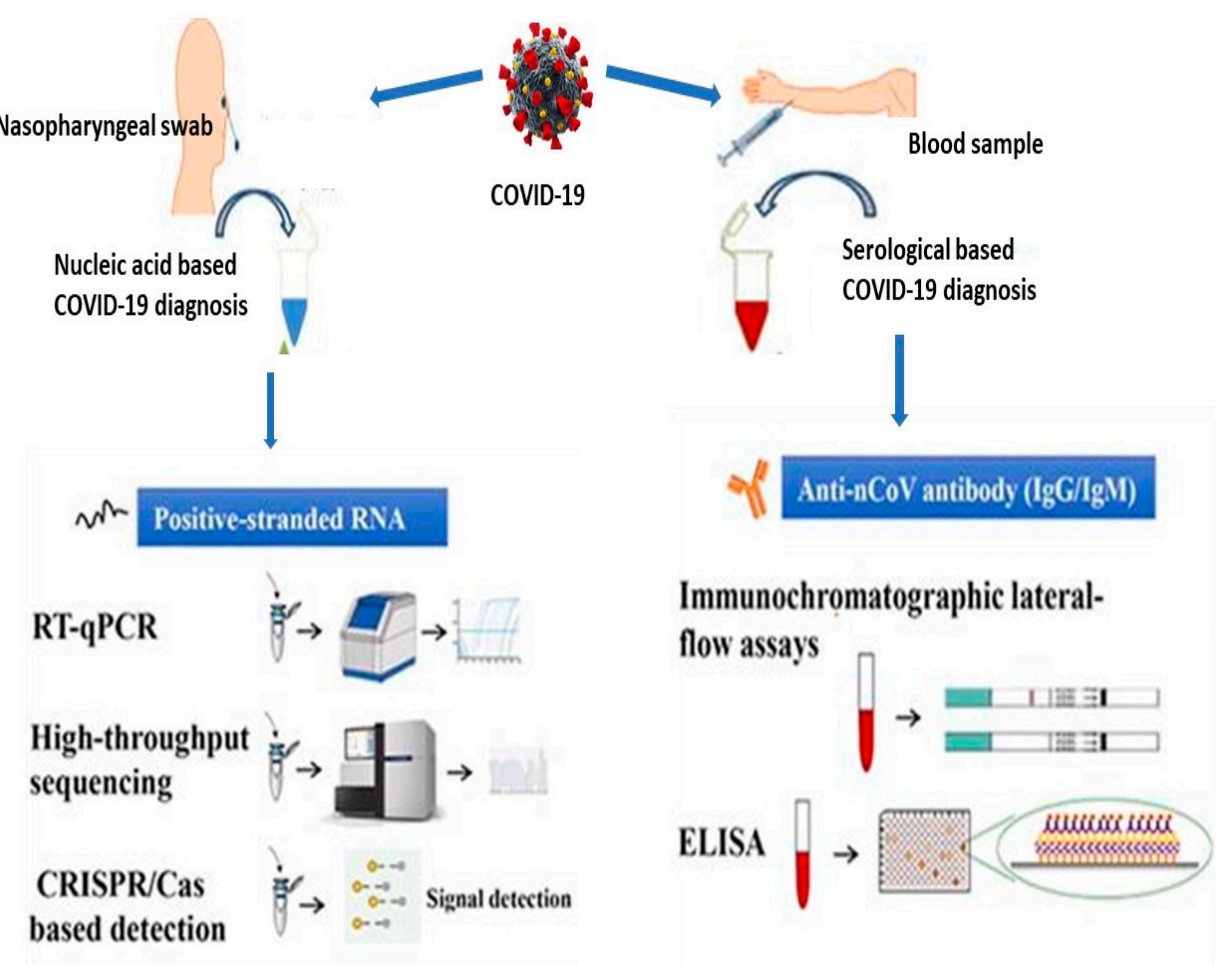

**Figure 2.** Schematic presentation of different molecular detection approaches and sample collection techniques for COVID-19. Adapted with permission from Ref. [64]. Copyright 2020, Frontiers.

Isothermal Amplification Techniques with Point-of-Care (POC) Potential

The analytical applications of CRISPR-based diagnostics techniques involved the use of mainly Cas12 and Cas13 proteins, which are responsible for targeting ssDNA and ssRNA, respectively. The assay involving Cas12 needs mediation of Protospacer Adjacent Motif (PAM) to promote the relaxation of dsDNA targets and facilitate the interactive binding between the DNA and crRNA, inducing the cleavage of Cas12 to ssDNA, whereas Cas13 cleavage to ssRNA does not require PAM sequence mediation. However, the synergistic function of CRISPR techniques (using both Cas12 and Cas13) with isothermal amplification approaches has been demonstrated to outperform SARS-CoV-2 RNA detection. The substrate is quenched and fluorophore-leveled ssDNA or ssRNA, with their separation detectable in real-time after cleavage use or under LED light intervention. Furthermore, when the substrate is leveled with fluorescein amide (FAM) and biotin, strip detection is possible [73]. When the sample was identified, the appearance of a single red band due to AuNPs interacting with streptavidin (in the absence of SARS-CoV-2 RNA) confirmed a negative test for COVID-19. While the positive sample produces the second red band as a result of free FAM interacting with AuNPs after the activated Cas protein is cleaved with signal reporters [71].

High-Throughput Sequencing Technology

High-throughput sequencing technology is one of the most accurate methods for determining genomic variation in pathogenic organisms. The Chinese Center for Disease Control and Prevention used this method to identify genetic sequence similarities between

SARS-CoV-2 and SARS-CoV viruses. The China National Medical Products Administration developed the DNBSEQ T7 (kit and software) to determine the genomic sequence of SARS-CoV-2 and related viruses in 20 h [74]. It can generate more than 100 million data outputs in each cycle and complete sample analysis in 50–200 cycles. The nanopore sequencing technology allows for real-time and complete sequencing analysis of the 2019 nCoV using an electrical signal, with no enzymatic intervention required. In 2020, the Hangzhou Center for Disease Control and Prevention used nanopore technology to determine the genomic sequence of COVID-19. The Nanopore technology expects to provide an alternative for monitoring potential COVID-19 mutations [75].

Isothermal Temperature Nucleic Acid Amplification Technology and Recombinase Aided Amplification (RAA)

Jiangsu Qitian Gene Biotechnology Co., Ltd. and the Chinese Center for Disease Control used RAA technology to develop a novel isothermal amplification kit for the rapid detection of SARS-CoV-2 nucleic acid, and the kit analyzes the 2019 nCoV sample within 8 to 15 min. The use of this kit as a standard kit for diagnosing COVID-19-suspected patients in China is being considered [76].

Nucleic Acid Mass Spectrometry

Nucleic acid mass spectrometry is a new type of advanced technology that uses Assisted Laser Desorption Ionization to detect known mutations in respiratory viruses such as SARS-CoV-2. The technology can amplify approximately 20–50 PCRs and detect a dozen pathogens at a time. In practice, the kit can analyze 1504 samples in a single day. Another advantage of this kit is its low cost and sensitivity, though the resolution needs to be improved [71].

### 3.1.2. Antibody-Based Sensing

Antigen identification is a potential tool for pathogen detection in both recovered and asymptomatic patients, which serves as essential means of accurate determination of the disease's prevalence. Enzymatic Link Immunosorbent Assay (ELISA) serology testing is considered among the forefront measures used for the diagnosis of SARS-CoV-2, and it is more advance than RT-PCR tests that can only detect the pathogen at the period of infection [59,77]. However, developing a point-of-care biosensor for rapid COVID-19 diagnosis based on the detection of anti-SARS-CoV-2 antibodies (IgM and IgG) is not appropriate. Because these antibodies take two weeks to be produced after infection, there may be a subset of patients who do not produce antibodies after infection with COVID-19. A sensor of this type could be used to assess the efficacy of SARS-CoV-2 vaccines [78]. Meanwhile, several pharmaceutical companies have developed many antibody-based testing protocols to detect COVID-19, most of which have obtained FDA approval.

### 3.2. Biosensors and Signal Detection Methods for COVID-19 Diagnosis

The use of nanomaterials, particularly AuNPs, AgNPs, and quantum dots as labeling agents (nano-labeling), amplifies the signal detection efficiency of biosensors and their use of special ultra-sensitive biosensing techniques [79]. Various forms of nanobiosensors have been developed from textiles, fabric, carbon, and thread to improve the existing biosensors' performance. On the one hand, the application of these biosensors and antibodies (AuNP-Ab) could establish high sensitivity in the pathogen's detection process. As such, this technology could potentially be used to detect IgG and IgM of COVID-19 in a suspected sample [80,81]. In addition to graphene and black phosphorus-based biosensors, several other biosensors have been developed from 2-D materials, and they are useful for point-of-care diagnosis of the COVID-19 virus [82,83]. Table 2 summarizes the various biosensors and their principles of working.

**Table 2.** Summary of various nanobiosensors for detection of SARS-CoV-2 and related viruses.

| Biosensor | Principle | Targeted Component | Detection Time | Ref. |
|---|---|---|---|---|
| Optical fiber-based biosensor | Act by the measurement of absorbance, refractive Index, fluorescence, and chemiluminescence | Bio-receptor | Immediate | [84,85] |
| Indirect immunofluorescence biosensors | Determination of antibody and antigen substrate against infectious disease | Epithelial sodium channel | Several hours | [86] |
| Nanowire biosensors | Field-effect transistor application | DNA sequence and specific protein | 44–46 min | [87] |
| RT-LAMP-VF | Isothermal amplification labeled using fluorescence iso-thiocyanate | N-gene of the MERS-CoV | 30 min | [88] |
| Amperometric biosensor | Production of a potential or current energy, which is proportional to the concentration of the detected substance | Glucose concentration | 15 min | [89] |
| Piezoelectric biosensor | Acoustics (sound vibration) | Alters crystal surface frequency | 10 min | [90] |
| Conductormetric biosensor | Cheap thin-film standard technology | Enzyme, whole-cell and DNA | Short time | [91] |
| Localized surface plasmon resonance | Electromagnetic incident light excited the electrons of metal nanoparticles to oscillate | Nucleic Acid | NA | [92] |
| Lanthanide-Doped Nanoparticle-based Lateral Flow Immunoassay | Detection Antibody Conjugated with AuNP | Phosphoprotein, Specific IgM anti-P. jirovecii antibody | 10 min | [93] |
| DETECTOR (Lateral Flow detection) | Amplification of viral genetic material | Nucleic Acid | 45 min | [76] |
| CRISPER/Cas-12 a based detection with the naked eye | Generation of green fluorescence in blue light observable with the naked eye | Nucleic Acid | 45 min | [94] |
| Field-Effect Transistor-based Biosensor | The semiconductor devices work by the change in potential differences that result in the sensitivity of the indicator | Spike protein | Few minutes | [95] |
| Reverse Transcription Loop-Mediated Isothermal Amplification (RT-LAMP) | Synthesize and amplify cDNA from template RNA to detect them using LAMP technology | Nucleic Acid | 30–45 min | [96,97] |
| RT-LAMP + lateral flow | The assays containing positive COVID-19 produce a green light using Visual detection reagent | ORF 1a/b | 60 min | [98] |

Zhang, L.L. et al. have developed SPGE "screen-printed graphene electrodes" useful for point-care diagnostics due to their high sensitivity and excellent electrochemical properties. The GSPE biosensor was developed by incorporating or substituting the graphene nanoparticle into the conventional SPCE "screen-printed carbon electrodes" [99,100]. The samples are collected either from the blood or nasopharyngeal swab, and the method of detection involved the quantification of colorimetric loop-mediated isothermal amplification (LAMP) resulting from the changes in cathodic current following the interaction between SPGE and amplicons [101]. Point-of-care biosensors, such as chip-based biosensors [102], paper-based biosensors [103], film-based biosensors [104], thread-based biosensors, [105] graphene-based biosensors [106], and black phosphorus-based biosensors [93], have played a significant role in detecting and alleviating the spread of SARS-CoV-2 pandemic infec-

tion. Smartphone analysis could now be used for rapid point-of-care testing of pathogenic diseases, including COVID-19. As such, this would promote a global system of healthcare management [107]. These applications are presented schematically in Figure 3.

### 3.2.1. Optical Biosensors

Optical biosensors refer to modern analytical technology that uses optical transducers for the bio-recognition of organic molecules. The fluorophore molecules attached to the target molecules enhance the optical biosensor sensing system, measuring the bimolecular analytic concentration carried out by the analytical, absorption, fluorescence, and phosphorescence chemo-luminescence and reflection [108–111]. Most optical biosensors were developed using the concept of plasmonics; the optical sensors consist of an optoelectronic transducer, light and detector. The optical biosensors have been in clinical use for viral detection since the 1990s and are currently being modified to detect COVID-19. For instance, LSPR and plasmonic photothermal effects have been employed in the diagnosis of SARS-CoV-19 [112].

In a recent development, surface plasmon resonance (SPR) in metallic nanoparticles has been developed with high sensitivity and specificity to target molecules. M. Alafeef et al. investigated the development of a biosensor for colorimetric detection of SARS-CoV-2 (via the naked eye) by targeting the viral N-protein with plasmonic AuNP conjugated with antisense oligonucleotides (ASOs). The sensor demonstrated high sensitivity and rapid detection of SARS-CoV-2 samples [113]. However, the Swiss Federal Laboratory has applied this technology and developed a biosensor that can precisely detect the presence of COVID-19 in the sample and its concentration in air aerosols. The local-SPR biosensor was developed using two-dimensional gold nano-islands (AuNI) containing functional DNA receptors capable of detecting unique SARS-CoV-2 RNA sequences. Despite some limitations in technology, the ability to re-identify the viral mutant gene proves to be more accurate and timely than RT-PCR [92,114].

Juxin Yin et al. have developed a simple digital biosensor that detects and quantifies nucleic acid in less than 45 min. It works with the lysis of the sample cells and binds the nucleic acid to the magnetic beads, followed by isothermal amplification and fluorescence imaging techniques. The authors suggested that this biosensor's application would be of great importance in the detection of SARS-CoV-2 [115,116]. Jin Jiao and his research team have successfully developed a biosensor that quickly detects the presence of COVID-19 within 10 min at a wide temperature range of 15–35 °C. The biosensor works by quantifying the fluorescence intensity produced from isothermal amplification of the SARS-CoV-2 RNA. The concentration of the targeted protein presence in the analyst is directly proportional to the biosensor's fluorescence intensity [117]. Presently, Djaileb et al. developed an optical biosensor based on Surface Plasmon Resonance (SPR), which was suggested to be an alternative to oropharynx swab sampling. The diluted human serum of the nM range is used to detect anti-SARS-CoV-2 nucleocapsid antibodies in less than 15 min [118]. Qiu et al. have designed a dual-function biosensor from PPT and LSPR sensing material. The device involves the use of 2D gold nanoislands (AuNIs) functionalized with cDNA receptors allowing it to precisely detect the presence of the SARS-CoV-2 genome in a concentration of about 0.22 pM. Hence, this technique could be explored in a clinical setting for the diagnosis of SARS-CoV-2 [92].

### Lab-on-a-Chip Biosensors

Chip-based biosensors are among the newer generated miniaturized devices incorporating either polydimethylsiloxane (PDMS) or polymethyl methacrylate (PMMA) and used for the rapid detection of high-precision pathogenic infections using the minute quantity of samples [106,119]. They work via the application of molecular biotechnology and microfluidics systems in which a single drop of blood is sufficient to undergo several biochemical reactions with different biomolecules of antigens, antibodies, and oligonucleotides [120]. Its widespread application owing to its reliability and ease of handling, in addition to cost-

effectiveness and high biocompatibility, has attracted the attention of scientists [121,122] toward the widespread use of it as a point-of-care diagnostic device for the rapid detection of pathogenic organisms [123]. As a result, such biosensors could be designed to detect COVID-19 viruses in the future [107].

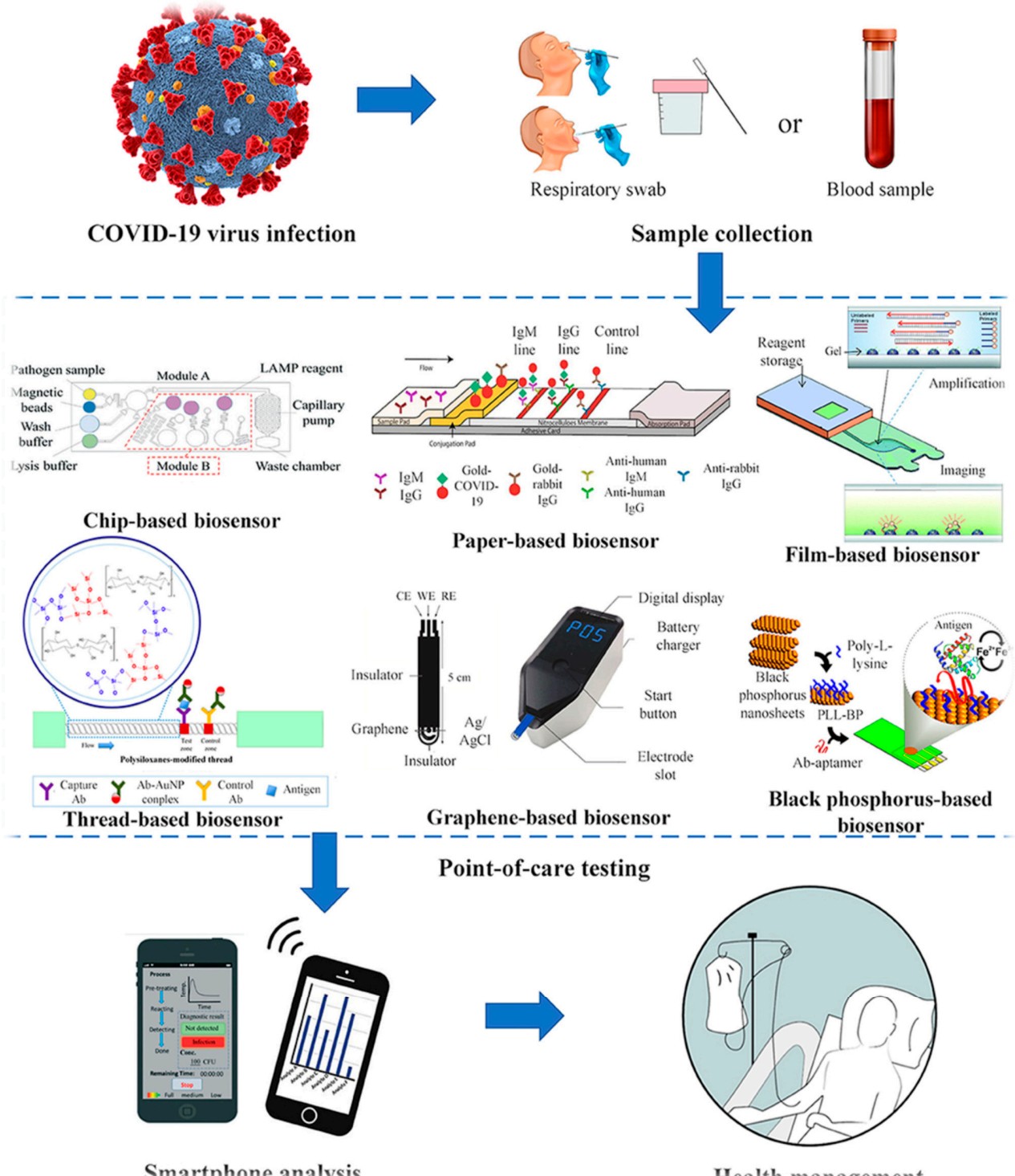

**Figure 3.** Point-of-care biosensors for detection of COVID-19. Reprinted with the permission of Ref. [107]. Copyright 2020, Frontiers.

Paper-Based Biosensors

Paper-based and lateral flow test strip biosensors are the most widely used point-of-care tests to detect COVID-19 immunoglobulins (IgG and IgM) since they produce colorimetric signals, are easily fabricated, and have low running costs [124,125]. Here, the lateral flow test strip comprises three zones, including a sample pad (for sample dropping), a conjugated pad (a region containing AuNPs- COVID-19 antigen followed by gold-rabbit IgG) and a nitrocellulose membrane. A region that contains the test lines, namely, the control line, IgG test line, and IgM test line, is coated with goat anti-rabbit IgG, anti-human IgG, and anti-human IgM, respectively, in addition to the absorbent pad. After adding the infected sample, an observable red color is produced at the strip's control line due to the reaction between the IgG gold-rabbit and the IgG anti-rabbit, followed by either positive IgM and negative IgG or positive of both IgG and IgM. This result indicates an acute infection, whereas chronic infection is indicated by observing positive IgG and negative IgM at the stripline [123,126,127].

This concept of 3-D paper-microfluidic biosensors is interesting and has been widely studied. The presence of pathogens is realized by colorimetric detection, which occurs when metallic ions of calcium, magnesium, and silver, etc., form a stable complex with pathogenic double-stranded DNA amplicons (or sodium sulfite and fuchsin) displaying a fluorescent signal under ultraviolet radiation [128,129]. Biosensors are made up of sample, reaction, and detection zones. Upon addition of the sample, the sample is prevented from escaping by covering it with the adhesive film, and the strip is inverted, allowing the sample to move into the reaction zone and be mixed with the hydrochloric acid and sodium sulfite injected. The reaction results in observable color changes of the fuchsia stained lines in the detection zone following a heat induction to the biosensor at 65 °C for 30 min. The use of these biosensors could facilitate the detection of SARS-CoV-2 infections at the point of care [107,130]. The paper-based electrochemical biosensor designed by Singhal et al. was suggested to have high efficiency in chikun-gunya virus detection. A simple modification of this device to target SARS-CoV2 genomic sequences using AuNPs associated with magnetic NPs ($Fe_2O_4$) could be economical and clinically advantageous [131,132].

Graphene-Based Biosensors

Graphene is one of the essential nanomaterials with unique properties. Having 'no inter-molecular bandgap makes it the best conductor with high optical transparency. It is stronger than the diamond, more elastic than rubber, and lighter than aluminum [133]. For the diagnosis of various medical conditions, the use of phosphorene-based nanobiosensors has been implemented [134,135]. Hence, graphene and black-phosphorous "BP" possess excellent electrochemical and redox properties in addition to their sensitivity and high se-lectivity [136], and BP-nanosheet coated with poly-L-lysine (PLL) enhances its functionality with anti-Ab-aptamers [137]. These biosensors allow the rapid detection of SARS-CoV-2 immunoglobulins in blood samples [138]. Y. Chen et al. developed a FET biosensor using rGO for the detection of the Ebola virus. In the device, the electrodes of $Al_2O_3$ and AuNPs were immobilized by anti-Ebola antibodies allowing the real-time detection of the Ebola virus's Zaire strain in a sample of low concentration of about 1 ng/mL [139]. Lin-Xia Fang has modified the carbon electrode-based biosensor by capping the electrode with AuNPs and graphene that is immobilized with aptamer. The sensor's high sensitivity and low level in the detection of avian influenza (H5N1) virus ($5.2 \times 10$–14 M) is due to the high accommodation of AuNPs in the large surface area of graphene [140]. Fereshteh et al. mod-ified the previously designed carbon electrode-based biosensor by fabricating the glassy carbon electrode with porous rGO and $MoS_2$ and then immobilized it with a specifically designed aptamer for detection of human papillomavirus (HPV)-16 L1. Upon validation, the modification increased the sensitivity and low detection capacity of the sensor up to 0.1 ng. $mL^{-1}$ [141]. Sengupta et al. reported that Graphene-FETs based biosensors are the cheapest and simplest biosensors to be fabricated, but they only suffer from the drawback

of current leakage, resulting in low sensitivity. The proper bandgap could render the device more sensitive and accurate [142].

3.2.2. Electrochemical Nanobiosensors

These molecular sensing devices use electrochemical transducers to detect the content of biological analytes and convert the content information into quantifiable signals. The signals are generated by biological reactions that occur on the surface of the electrode and are detected using amperometric, impedimetric, or potentiometric methods [143]. The three electrodes contained within the electrochemical biosensors, which include the counter, working, and reference electrodes, can perform a variety of functions when modified. Working electrodes, for example, can be modified with viral-specific antibodies or genomes to allow for selective detection. In 1962, Clark and Lyons invented the first electrochemical biosensor to detect the concentration of glucose in the analytic sample [144]. The amount of oxygen consumed by the oxygen electrode of the device is proportional to the amount of glucose present to elicit the signal. The application of nanotechnology has transformed the functionality of electrochemical biosensors, improving their sensitivity and efficiency in detecting bacteria and viruses. According to Pingarrón et al., incorporating AuNPs into an electrochemical biosensor increased its sensitivity and specificity for detecting viral genomes [144,145].

Field-Effect Transistor (FET)

The FET is one of the methods used in electrochemical biosensors for pathogen detection. It consists of three-terminal electrodes, namely the source, drain, and gate electrodes. During the analytic recognition, the bio-receptors mobilize to the gate electrodes and an electric potential is applied, causing a change in conductivity across the drain and source electrodes that are connected to the gate [146]. Joshi et al. modified an electrochemical immunosensor for the detection of influenza A virus (H1N1 subtype) by using thermodynamically decomposed rGO (TrGO). Similarly, Mikua et al. created an electro-immunosensor by modifying a gold electrode with a 0.01 mM dipyrromethene-cupper II (DPM-Cu) and 1 mM 4-Mercaptobutanol solution (MBT). The viral detection was made possible by the formation of a coordinate bond between the histidine tag H1 subtype of human influenza (His6-H1 HA) and Cu (II) [147]. Miniaturized electrochemical biosensors are required during the COVID-19 pandemic era to provide a point-of-care diagnosis because of their ease of fabrication, high sensitivity, and low cost in the detection of the SARS-CoV-2 genome and other proteins [148].

Tripathy et al. revised their miniaturized electrochemical biosensor, which had previously been designed for the detection of other viruses to detect SARS-CoV-2. The researchers coat the titanium-working electrode with AuNPs and use the gold-thiol bond to immobilize the ssDNA probe, allowing them to detect the SARS-CoV-2 protein or genome specifically [149]. Seo et al. created an FET-based biosensor by incorporating graphene sheets into the standard electrode and using 1-pyrene butyric acid N-hydroxysuccinimide ester (PBASE) to immobilize anti-spike protein antibodies (SpAb). The author uses nasopharyngeal swab samples collected from COVID-19-infected patients to validate the device's performance [95]. M. Alafeef et al. reported the development of a paper-based electrochemical biosensor for the detection of COVID-19 N-protein. The sensor was found to accurately diagnose COVID-19-positive samples in less than 5 min after clinical validation. It was also demonstrated that a sensor could be used to detect mutant genes using ssDNA-conjugated AuNPs [150].

Furthermore, scientists from India's National Institute of Animal Biotechnology (Mahari et al.) have developed a biosensor (eCovSens) that can detect COVID-19 in less than 30 min. The device was created by immobilizing anti-s-protein antibodies (SpAb) on AuNPs and a Fluorine Doped Tin Oxide (FTO) electrode that was coated with AuNPs. Therefore, eCovSens could have the potential for clinical use as point-of-care testing for COVID-19 [151]. Electrochemical data are generated either using voltammetry techniques

that include differential pulse voltammetry (DPV), amperometry, cyclic voltammetry (CV), and square-wave voltammetry (SWV) or using potentiometric techniques via electrochemical impedance spectroscopy (EIS) [152].

### 3.3. POC Diagnostic of COVID-19 Supported by AI and IoT

Incorporating Artificial Intelligence (AI) and the Internet of things (IoT) concepts into the POC sensing technique could enable the early detection of the COVID-19 pathogen and the management of bioinformatics data. The incorporation of AI in data collection and analysis of COVID-19 cases is essential for intelligent healthcare delivery and predicting the infection's trajectory. Exploring these innovations in the form of smartphones has been the most effective approach for understanding the disease spread patterns, prevention, and control, as a larger percentage of the population owns a smartphone. Such user-friendly AI-supported nano-enabled electrochemical approaches are essential in optimizing the SARS-CoV-2 sensing system for POC intelligent diagnosis of COVID-19 in a personalized manner [153,154].

## 4. Inactivation of SARS-CoV-2 Transmission Using High-Performance Nanosystems

According to current literature, nanoparticles of different metals, metal oxides and carbon-based are extremely promising for killing viruses through different mechanisms [155,156], as schematically presented in Figure 4.

### 4.1. Blockage of Cellular Infiltration

The blockage of cellular attachment of S-protein and ACE2 receptors of host cells could be a potential target to prevent COVID-19 infection [157]. The technique of using nanoparticles to fight against SARS-CoV-2 may include pathways that will influence the virus' entry into the host cell before its inactivation. Blocking the viral surface proteins lead to its inactivation, so selective nanoparticles, unique to proteins expressed in viruses, may minimize viral internalization. Metallic NPs have shown great affinity for blocking the process of viral cellular entry [158]. AuNP-based nanomaterials have strong potential for use in anti-novel coronavirus [159]. AgNP prevents surface contamination of COVID-19, CQDs and AuNPs could bind to many antigens due to their large surface area and inhibit both cellular entry and surface contamination [160]. Rabiee N et al., reported the affinity of CQDs in conjugate with boronic acid-NPs to prevent the cellular interaction of COVID–19 S-proteins and ACE2 enzyme of the host cell, deterring the virus cellular integration more significantly [161]. Moreover, an acid-functionalized multi-walled CNT composed of photo-activated molecules was reported to have inactivated the viruses and prevented their cellular entry [162].

### 4.2. Inhibition of Cellular Replication

AgNPs have been used for antiviral therapy against SARS-CoV-2, it works by inhibiting the virus's replication process and preventing its cellular entry while reducing the cellular PH [163]. Copper nanoparticles (CuNPs) have been reported to have shown potent antiviral properties by blocking papain-like protease-2, which in turn, disrupts its replication processes [164]. In addition, potential interactions between the negative RNA chain of COVID-19 and quantum dot NPs have been reported to block the COVID-19 –ACE2 binding site and affect genomic replication [165]. MC Sportelli's study revealed that AgNPs have a potential inhibitory effect on viral replication. Similarly, AuNPs conjugated with biocompatible polymer demonstrated antiviral activity against HIV-1 and specific influenza virus sub-types (e.g., H1N1, H3N2, H5N1) [166]. QDs utilize its cationic surface charge and interfere with the RNA replication of SARS-CoV-2 by inducing the accumulation of ROS in the virus [167]. In addition, the interaction between nanoparticles/materials and components of the SARS CoV-2 virus at the molecular level resulted in learning about several significant effects against the inhibition of viral cell entry, genomic replication, and eventual induction of high immunogenic reactions that can facilitate vaccine development.

Several NPs, such as copper, zinc, silver, silicon, and gold, have been studied for COVID-19 vaccine and drug development through various mechanisms [168]. Raha, S. et al. conducted a study on the efficacy of glutathione-Ag2S-NPs on Porcine Epidemic Diarrhea Virus (PEDV), a SARS-COV-related family. The research has shown a significant inhibitory effect of glutathione-Ag2S-NPs on different stages of the COVID-19 life cycle, such as negative-stranded RNA synthesis that induces a robust immune response, which helps in neutralizing the cellular viral antigen. The author used carbon curcumin dots NPs and found similar results, indicating the possibility of using these capped nanomaterials in COVID-19 management [169]. The description of interaction mechanisms of NPs with COVD-19 has been summarized in Table 3.

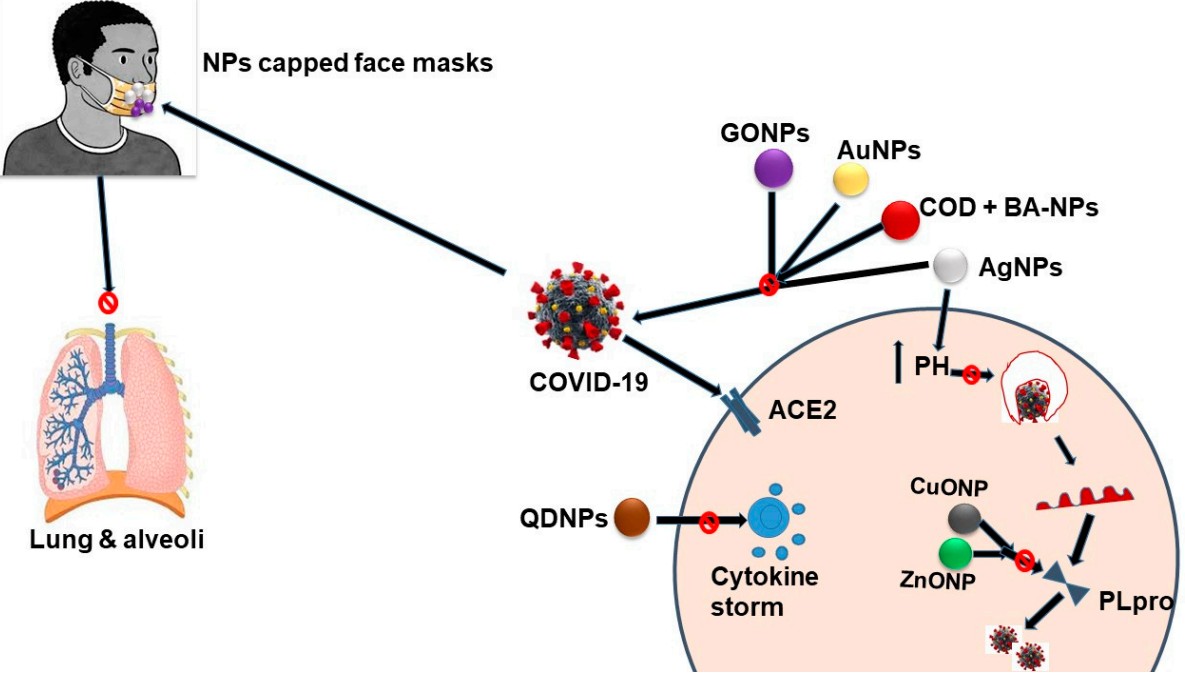

**Figure 4.** The role of nanomaterials for prevention and inactivation of COVID-19 infection. CuONP is a copper oxide nanoparticle, ZnONP is a zinc oxide nanoparticle, cBA-NP is a boric acid nanoparticle, GONP is a graphene oxide nanoparticle, COD-NP is a carbon oxide nanoparticle, AuNP is a gold nanoparticle, AgNP is sliver nanoparticle, and QDs is quantum dots.

**Table 3.** Nanoparticles and their mechanism of targeting and detecting SARS-CoV-related viruses.

| Nanomaterials | The Mechanism of Action | Remarks | Reference |
|---|---|---|---|
| Silver nanoparticles (Ag+) | AgNPs act by promoting and forming a process for Ag+ coordinating to the iodine probe base on irreversible tandem ring-opening in which the color and fluorescence changed | It has a veridical effect when used in a facial mask, and it reduces the titter of COVID-19 (SARS-CoV-2) to zero | [170] |
| Gold nanoparticles (AU) | Detect MASS-COV using double-stranded DNA by calorimeter assay | Eosinophilic infiltration in lungs | [171] |
| Zinc oxide (ZnO) nanoparticles | Lowe bacterial and viral detection capacity in patients with minimal viral load | Limit the cytokine storm in COVID-19 and limit the damage to the tissue | [172] |
| Zinc sulfate (ZnSNps), nanoparticles | Substrate specific preferences of SARS-CoV and SARS-CoV-2, but mechanism remains unclear | Increases the alcohol content beyond the comfort zone in yeast pushing | [173] |

**Table 3.** *Cont.*

| Nanomaterials | The Mechanism of Action | Remarks | Reference |
|---|---|---|---|
| Platinum nanoparticles | The nanocomposite of platinum characterizes by a transmission electron microscope in which the average particle size of 2.2 nm $\pm$ 0.6 | Very low tolerable drug payload in antibody drugs conjugated | [174] |
| Magnetics nanoparticles | Detection of virus and virus genome in immunoassay due to the alternation of magnetization with changing electrical resistance from the high volts into low volt | Its application limited to in vivo tests that demonstrate the efficacy of graphene oxide | [175] |
| Silica-coated magnetic nanoparticles | Work in sandwich hybridization assay for cDNA target detection with a limit value of $2.0 \times 10^3$ copies within 6 h | Do not have an antifungal effect; it is limited only to the microorganism's circulation in the air | [176] |
| Cadmium sulfide (CdS)-NPs | In the ambient temperature, the core-shell nanowire of CdS possesses superior intense green luminescence emission, which demonstrates charge performing carrier's recombination inside the CdS core | The detection limit in the fluorescence intensity range is $1.2 \times 10^{-11}$ | [177] |
| Lipid-based NPs | It enhances drug delivery in a controlled manner with high drug loading capacity | Because of its crystallization, it has the possibility of drug expulsion upon storage and low drug loading efficiency | [178] |
| Solid lipid NPs | It decreases the effect of first-pass metabolism and hence enhanced tissue distribution of drugs | | |
| Nanoemulsions (NEs), | It works by increasing bioavailability and lymphatic uptake of some drugs such as saquinavir or indinavir | Instability, low solubilizing capacity for high-melting substance | [65] |
| Polymer-based NPs | It improves the pharmacokinetic parameters of the drug preventing its early degradation and side effects | Biphasic drug release | [179] |
| Dendrimers | They have excellent cellular uptake with longer circulation times, and inhibiting viral entry fusion has been successful in treating HIV and HSV2 infections | Low aqueous solubility and high nonspecific toxicity | [180] |

### 4.3. Surface Sterilization/Personal Protection

Since coronavirus was declared a global emergency, different strategies have been adopted to eradicate all such measures. However, scientists have proposed the possibility of transmission of infection from the aerosols and droplets of infected persons when they coughed, sneezed, or spoke. As such, the use of a face mask and personal protective clothing can drastically reduce these viral particles' transmission [181]. Shuo Feng et al. showed that a populace who adopted the tradition of using face masks has less prevalence of infection than those who neglected it [182].

According to the existing literature, copper, silver, and zinc nanomaterials have been explored as potential microbial agents against many pathogenic organisms. These materials may be beneficial for the prevention of COVID-19. Additionally, copper salt nanoparticles have a potent antiviral effect that could be used in the development of PPE [183]. The combination of copper and quaternary ammonium nanoparticles provides a synergistic antiviral effect that could effectively reduce the spread of SARS-CoV-2 [184]. The combined research team of Z.E.N. Graphene Solutions Ltd. (TSXV: Z.E.N.) and Graphene Composites Ltd. (G.C.) has developed a new graphene-nanoparticle ink. The ink reported having a potential antiviral property used as a PPE to block and kill SARS-CoV-2 viruses. They also synergized the antiviral effect of graphene oxide ink by combining it with synthetic silver nanoparticles. The trial is still underway at Western University Canada; once safety has

been established, a breakthrough in viral transmission would be achieved by incorporating ink into a strong PPE fabric [185,186].

### 4.4. Reusable Face Masks

The increase in demand for face masks has made the available cover insufficient because it can only be used once to prevent cross-infection by airborne pathogens that remain in the masks' fibers. Therefore, it is necessary to develop these masks' reusability and strengthen their protection against COVID-19 pathogens by capping them with different nanoparticles/materials [187].

Jan Mikulicz 1897 was the first to record the use of medical masks [180]. As stated earlier, coronavirus transmission is through aerosol droplets from coughing, sneezing, breathing, or speaking of the infected person, remaining in the air as suspension for a distance of 2 m. However, wearing a surgical face mask will significantly reduce the spread of viral particles [181]. Kaushik et al. explored the use of smart and novel biomaterial for strengthening the antiviral protection of face masks and minimizing the environmental transmission of SARS-CoV-2 [188].

In addition, silver-based nanocomposites owed the highest antimicrobial protection affinity through unknown mechanisms. Although, there is evidence that some theories suggest that positively charged silver ions could disrupt pathogenic membranes that impair their life cycles, causing immediate cell death without causing any considerable side effects [189,190]. The Leung, P.H et al. study has shown that $AgNO_3$ and $TiO_2$ nanoparticles possess 100% protection affinity against *E. coli* and *S. aureus* bacteria when acting synergistically on the fiber of the respiratory mask [191]. Hiragond et al. coated a commercial face mask with varying concentrations of AgNPs and found the highest antimicrobial protection affinity at a concentration of 100 ppm AgNPs. The author suggested that the procedure could be a breakthrough in forming safe, reusable, and effective face masks [192]. There is a need to conduct more research to establish the safety of this method for practical applications. The desirable properties of graphene, which include high UV and fire resistance, in addition to high conductivity and low molecular weight, have embraced its application in personal protection against infections, and this could significantly help in making PPE effective against COVID-19 [193]. A report from Ramaiah et al. proved that graphene-based face masks such as polygreen and nanene have 95% air filtration efficiency against airborne bacterial and viral infections [194].

### 4.5. Photodynamic Approach to Eradicating SARS-CoV-2 Virus

The clinical application of photodynamic therapy PDT remains a potential approach for the treatment of oncological disorders. Its antiviral effect was proved in the 1970s. The PDT provides a unique approach to mitigating the SARS-CoV-2 pandemic infections [195]. The PDT damages the viral proteins nucleic acids and possibly lipids, leading to virus inactivation, which occurs following the interaction between the viral proteins and reactive oxygen species (ROS) generated by the excited photosensitive agents known as photosensitizers PSs. Unfortunately, the clinical application of PDT is associated with many challenges, including low penetration, poor specificity hydrophobicity and aggregate formation in aqueous media [196]. To overcome these hindrances, Lim et al. have proposed the incorporation of NPs in PDT. The author manipulates sodium yttrium fluoride (NaYF4) up-conversion NPs (UCNs) by coating them with polyethyleneimine (PEI). As a result, the UCNs were found to be effective against many viruses, such as Dengue-type 2 and adenovirus-type 5 [197].

In addition, several 2D nanomaterials, such as MXenes, black phosphorus, graphene, graphitic carbon nitride, and tungsten disulfide, were employed in PDT for chemotherapy. Amongst this, graphene and fullerene were proven to elicit excellent antiviral activity against influenza A vesicular stomatitis virus (VSV), HSV-1, and HIV-1 among others. The MXenes, in particular, have shown desirable catalytic, optical, and electronic properties enabling them to acquire a wide range of theranostic and PDT applications. Therefore,

exploring nanomaterial-based PDT in the SARS-CoV-2 inactivation should be of great significance [198].

The mechanism of PS-virus inhibition involves the electrostatic interaction between the positive charge of PS and the negative charge of virus envelopes or nucleic acid. Therefore, enveloped viruses such as SARS-CoV-2 are significantly more likely to be cured by PDT than undeveloped viruses. The major target for PDT in SARS-CoV-2 structure is the spike protein (S1 and S2), as well as the nucleic acid. The PDT of SARS-CoV-2 via type-I process comprises the generation of free radicals by PS that interact with either nucleic acid or outer proteins of SARS-CoV-2, resulting in photo-oxidation of proteins, whereas the type-II process involves the interaction of singlet oxygen produced by PS and guanine residues of SARS-CoV-2, [199] as demonstrated in Figure 5.

Being a SARS-CoV-2 encapsulated virus makes it more prompt to lysing by ROS. This could be supported by Mie's theory, which suggested that "Metal nanoparticles are excellent photolytic agents can absorb light even with the smaller diameter than the wavelength of the incident UV light. Forming the decay of a surface plasmon polariton". Negrete, O et al. have reported the photocatalytic effect of TiO2 capped with AgNPs when the UV light of 400 nm (3.1 eV). Hence, the AgNPs could produce electrons continuously to the TiO2 conduction band even in the absence of UV light, cartelizing the formation of ROS to inactivate the encapsulated viruses, including SARS-CoV-2. Moreover, Svetlana K. et al. have demonstrated the photocatalytic effect of nanosized TiO2 (TNPs) using UV radiation in vitro. The study indicated that the TNPs could be an excellent photocatalytic agent for SAR-CoV-2 [200].

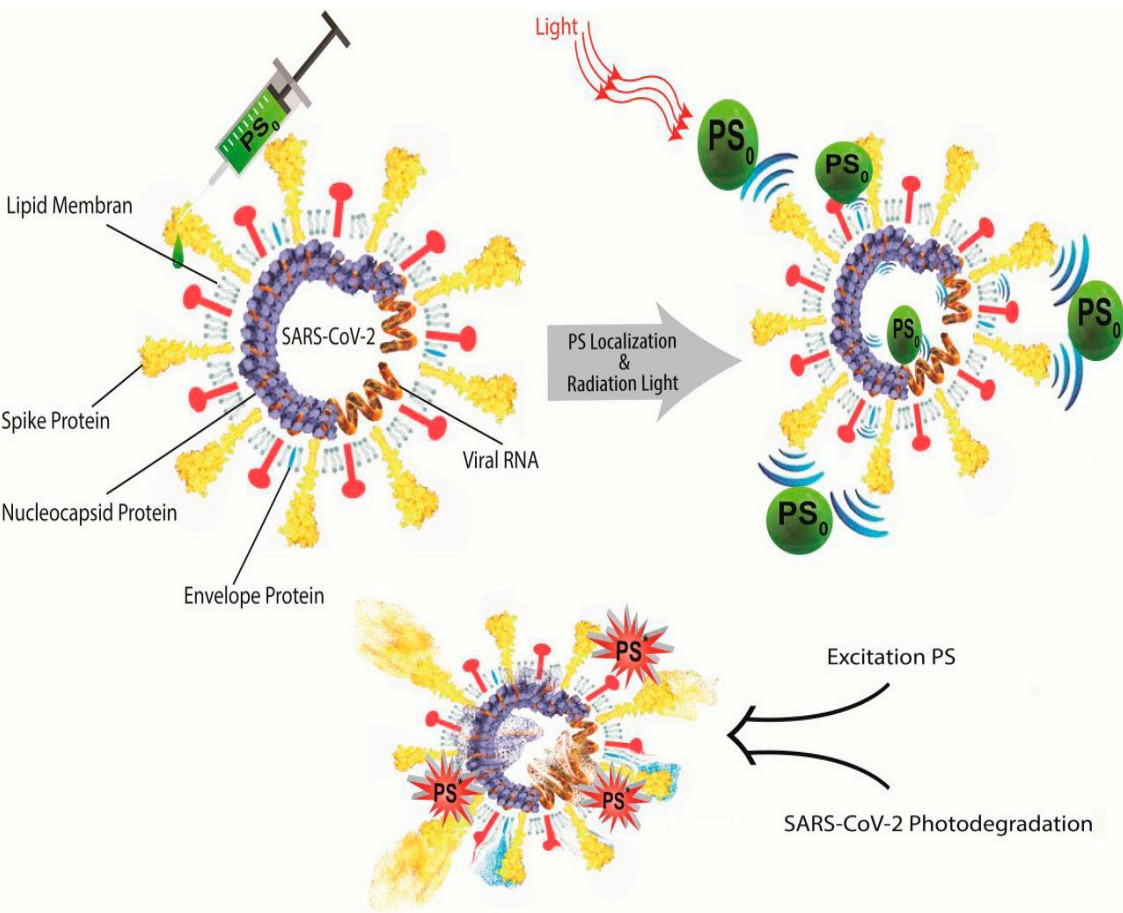

**Figure 5.** Schematic presentation of PDT protocol on the specific targets of SARS-CoV-2. This figure has been reprinted with permission from Ref. [199]. Copyrigt 2021, World Scientific.

### 5. Nanotechnology Based-Vaccines against COVID-19

The structural components of SARS-CoV-2 provide insight into the development of the vaccine against it. However, most SARS-CoV-2 vaccines' primary target was COVID-19 Spike protein, which stimulates and enhances the production of S-protein neutralizing antibodies [201]. The S-pike protein's trimeric form has a receptor-binding domain, enabling its entrance into the cell through ACE2 receptors. It is tedious to develop a SARS-COV-2 vaccine from biomolecules that mimic human disease in vivo because it is associated with a high level of immunogenicity and inefficiency [202]. Whereas the vaccine developed from natural SARS-COV2 is related to short-term immunity with a high tendency to re-infection [203]. The development of a human vaccine has taken on a new dimension as the use of DNA nucleic acid composition, RNA, viral vector, and recombinant protein has overtaken the traditional means of using live attenuated and inactivated viruses [201]. It has been reported that the vaccine developed from the specific viral antigenic protein subunits presents a minimal risk of adverse reactions compared to the vaccine produced from attenuated or inactivated viruses. Similar techniques were adopted in the production of MERS-CoV and SARS-CoV vaccines and such could be applied to SARS-CoV-2 [204,205]. Currently, most approved vaccines for COVID-19 and those in development have been designed to target S-protein because it may probably own non-neutralizing epitopes that limited the chances of occurrence of anti-SARS-CoV immune response by restricting the production of neutralizing epitopes [206–208].

Ting et Staroverov et al. reported that encapsulation of COVID-19 protein fragments in AuNP might be considered a potential antiviral agent for vaccine development [209]. Biocompatible hollow NPs can accelerate the production of effective and safe vaccines against emerging pathogenic coronavirus [210]. Kim et al. introduced ferritin-based NP assembly mediated by RNA as a novel molecular chaperone. They demonstrated that using this nanoparticle in MERS-CoV could stimulate CD4+ T cells, contributing to the formation of interferon-gamma (IFN-$\gamma$) and necrotic tumor factor-alpha (TNF-$\alpha$) [211]. Jung et al. developed an immunogenic nanovaccine using the "recombinant adenovirus serotype 5" encoding the "MERS-CoV" spike gene (Ad5/MERS) coated with NPs and administered the vaccine to one group and compared the result with the other group (vaccinated with NPs free coated vaccine). The result indicated that the group treated with nano vaccines showed higher antibody titers than the other group who received Ad5/MERS alone [212]. However, the viral inhibition assays conducted by DuT et al. showed that GONPs inhibited 25% of feline coronavirus (FCoV) and 23% of infections with Infectious Bursal Disease Virus (IBDV), while GO inhibited 16% of FCoV alone but did not show antiviral activity towards infection with IBDV. The GO combination with AgNPs can therefore multiply antiviral activity synergistically [213].

Sekimukai et al. reported 167 efficacy (GNPs) in developing a vaccine against SARS-CoV-2 as an antigen carrier that facilitates vaccines' delivery to host cells. Therefore, they are considered an essential adjuvant in the SARS-CoV vaccine that has been inactivated by ultraviolet radiation. Based on the available literature, an antigen-specific IgG response against SARS-CoV could be promoted by GNPs-adjuvant S-protein [214]. Nanocarriers can be loaded with different bioactive molecules such as antivirals and nucleic acids to target a particular disease or disorder, including the SARS-CoV-2 virus [215]. AuNP-coated anti-TGEV vaccine was tested against rabbits and mice groups. The vaccinated animals were found to have shown higher concentrations of IFN-$\gamma$ and more significant titers of neutralizing antibodies. Therefore, Au-NP conjugate could be essential for the development of the SARS-CoV-2 vaccine [216].

The use of nanomaterials and nanotechnology in vaccine development could improve vaccines' stability against enzymatic degradation, promote immunogenicity, and enhance the delivery of antigens to the cells [217,218]. However, StatNano has developed the idea of using virus-like nanoparticles to develop high immunogenic SARS-CoV-2 vaccines, and yet, the manufacturing process is quite challenging due to the high cost of production [219,220]. In addition, Morderna's vaccine was developed by encapsulating the S-protein mRNA with

Lipid Nanoparticles (LNP) that allow the body to synthesize the spike proteins internally and trigger the immune response to produce specific neutralizing antibodies [221]. Similarly, the Pfizer-BioNTech nano-vaccine candidate has also been developed with analogous technology achieving 95% efficacy in clinical trials [222]. Furthermore, the development of the Novavax nanocov-vaccine (NVX-CoV2373) has been reported to achieve high stability and immunogenicity. The delivery of these nano-mRNA vaccines could be possible via the oral route in the future [223]. However, 47 vaccines have been approved for use against COVID-19 in different countries, and some are illustrated in Table 4. According to a WHO report, 12.6 billion doses of COVID-19 vaccines were given globally, of which 4.9 billion people were fully vaccinated, accounting for 70 percent. Yet only 28 percent of older people and 27 percent of health workers in low-income countries received the primary dose of vaccine [224,225].

**Table 4.** Top successful COVID-19 vaccines approved and available for use outside of clinical trials. Sourced from the COVID-19 vaccine tracker.

| Sponsor | Vaccine | Description | Effectiveness | Status | Ref. |
|---|---|---|---|---|---|
| Moderna | mRNA-1273 | mRNA-based vaccine capped with LNPs | 94.1% | | |
| Oxford University AstraZeneca; IQVIA; and Serum Institute of India | AZD1222 | Replication-deficient viral vector-based vaccine (adenovirus from chimpanzees) | 70.4% | | |
| Pfizer, BioNTech | BNT162 | mRNA-based vaccine | 90% | | |
| CanSino and Biologics | Ad5-nCoV | Recombinant vaccine (adenovirus type 5 vector) | Encouraging | | |
| Sinovac | CoronaVac | Inactivated vaccine (formalin with alum adjuvant) | 92.4% | | |
| Bharat Biotech; National Institute of Virology | Covaxin | Inactivated vaccine | Encouraging | | |
| Johnson & Johnson | JNJ-78436735 (formerly Ad26.COV2.S) | Non-replicating viral vector | Hold on pending the outcome of ADR | Approved | [226] |
| Novavax | NVX-CoV2373 | Nanovaccine | Encouraging | | |
| Gamaleya Research Institute, Acellena Contract Drug Research and Development | Sputnik V | Non-replicating viral vector | 92% | | |
| Federal Budgetary Research Institution State Research Center of Virology and Biotechnology | EpiVacCorona | Peptide vaccine | Encouraging | | |
| Beijing Institute of Biological Products; China National Pharmaceutical Group (Sinopharm) | BBIBP-CorV | Inactivated vaccine | 86% | | |

## 6. Nano-Based Drug Delivery Systems for COVID-19 Management

The use of nanotechnology in drug and vaccine delivery has been considered a breakthrough alternative to overcome the current limitations of vaccine delivery. Nowadays, the delivery of small interfering RNA (siRNA) for the treatment of many diseases becomes successful when encapsulated in nanocarriers as applied in the Moderna COVID-19 vaccine [227]. The two main processes of delivering nanoparticle-based vaccine (NPb-V) include (i) encapsulating RNA/DNA or antigen within nanocarriers to protect them from

proteolytic degradation, allowing the direct delivery of vaccine into the nucleus of the target cell. (ii) The coating of antigen or RNA/DNA on the surface nanocarriers that allow them to mimic the virus [228]. For instance, the coating of SARS-CoV S-protein on the surface of nanocarriers could elicit a specific immune response and result in the formation of a specific antibody against it. Furthermore, the incorporation of purified human immunoglobulin (isolated from COVID-19-infected patients) on the top of nanocarriers could be a promising means of creating novel NPb-V against COVID-19 [229].

However, there are several promising nanocarriers used for vaccine delivery against SARS-CoV-related viruses, including COVID-19 [230].

The study of RNA-mediated therapies that naturally generated nano-vesicles known as exosomes has received high interest, owing to their ability to combine functional miR-NAs and transmit them to the target cells [231] Chi H. et al. reported that the in vivo vaccination of mice has shown a good prognosis. The vaccine was prepared from the SARS-CoV S protein incorporated into micellular NPs using Matrix M1 as an adjuvant excipient. The same approach was used by Novavax (NVX-CoV2373) in developing the COVID-19 vaccine [232]. Mao L. et al. obtained a similar result after using the same nanomaterial with aluminum adjuvant in vaccination against MERS-CoV [233]. Solid-lipid NPs (SLNs) have a great ability to mimic the virus and they could be easily modified to achieve their full potential, whereas the use of liposomes could be more adventurous in terms of vaccine development against respiratory viruses because it can be modified to resemble the virus particles and the vaccine can therefore be developed without the use of virus peptides. The same technology has been used to immunize mice against influenza upon intranasal administration [234]. In addition, the similarity of exosomes with the body cells makes it a double edge-sword for delivering both drugs and RNA or DNA-containing vaccines. Although the use of exosomes as NPb-V was limited due to the scarce amount to be fabricated [82]. Mazini L. et al. reported the safety and efficacy of intranasal administration of exosomes among severe COVID-19-infected patients [235].

The concept of encapsulating mRNA or siRNA in nanocarriers is a promising means of enhancing vaccine delivery and stability. For instance, Moderna's mRNA-1273 vaccine was developed by encapsulating mRNA in SLNPs, and it was found to be among the most successful COVID-19 vaccines [236].

## 7. Challenges and Perspectives

It is not naysaying that nanomaterials play a crucial role in tacking against COVID-19 pandemic; their applications include but are not limited to drug repurposing, vaccine development, diagnosis, spread mitigation and enhancement of the delivery efficiency of such drugs via various routes. Despite these advantages, several reports claim the association of nanomaterials (especially metallic NPs) with mutagenicity, toxicity and generation of free radicals that lead to cellular and tissue damage.

The identification of the SARS CoV-2 virus remains the main obstacle to its management amidst the pandemic outbreak. Rapid testing tools were necessary, especially for large cities and isolated locations with a high population density. This necessitated the application of cutting-edge bio-nanotechnology to the control of COVID-19 through the identification of appropriate bioactive functional systems with detecting capabilities. Perhaps the construction of nano-enabled miniaturized biosensors for SARS-CoV-2 virus protein detection exemplifies the concept of compartmentalization-based methodology. Bioinformatic primers are used in the application of point of care to correct infection progression based on race, gender and location. Although even the mutated forms of the COVID-19 virus, such as the delta strain, have caused a lot of problems, a recent vaccine approved for emergency use has had a lot of success. The major challenge with the COVID-19 virus is rapid mutation. The long-term goal is to develop a medicine that can help the immune system via nanotechnology.

Additionally, the problem of dose-dependency, biogenetic and biodegradability of NPs should be handled with caution. The clinical evaluation of nanoparticle-based in vitro

studies requires a critical assessment of biocompatibility, which could result in cellular irreversible damage. It is essential to understand the actual mechanism of cellular interaction of nanoparticles, which will pave the way to knowing how they trigger the irreparable damage to the cells, enzymes and virus structure. It is of great interest to design desirable NPs of high drug entrapment capacity and targeted release of the drug varying high bioavailability, distribution and low metabolism. However, the NPs can be employed to achieve both passive and active targeting due to their nanosized structure and ability to be functionalized with ligand and virus' specific antibodies, respectively.

## 8. Conclusions and Viewpoint

The world has recorded considerable success in the current fight against the pandemic of COVID-19 infection using innovative technologies. Among other techniques adapted, nanotechnology has provided a breakthrough in alleviating the spread of infection by manipulating nanostructures and nanomaterials in various ways. In this instance, the loading of the virus's RNA and specific proteins such as S-protein, M-protein or N-protein on the surface of nanocarriers have sped up the development of SAR-CoV-2 vaccines and allowed the design of nanosystems that mimic the virus' behavior, which in turn makes research on COVID-19 easier for scholars who do not have the facilities to access the natural SAR-CoV-2 virus. For example, the spread of the virus can be demonstrated using paramagnetic NPs because they can display CoV-like kinetics and target binding affinity. Finding the basic information on the nanostructure of the viral particle using computational stimulation of nano-interaction is necessary to establish the specific approaches on the road to discovering and developing effective vaccines, means of prevention and diagnosis of COVID-19. In addition, the concept could further enhance the vaccine delivery to the targeted site, minimize the side effects and sustain its circulation time, preventing the future reinfection and prophylactic use of the COVID-19 vaccine. The use of nanosystems has made a significant contribution to the development of recently approved vaccines against COVID-19 after the successful completion of a clinical trial. Of note, the first vaccine approved by the FDA (Morderna's vaccines) was developed with the concept of nanotechnology, and now 47 vaccines have been subsequently approved. Several studies predicted the possibility of intranasal delivery of such a vaccine, whereas the development of oral SARS-CoV-2 vaccines with the application of nanotechnology is also possible in the future. There is much hope for the development of sophisticated smartphone-based biosensors for rapid and early diagnosis of COVID-19 in the future. Scientists have further established evidence to strengthen the protection affinity and reusability of PPE by capping them with nanoparticles.

**Author Contributions:** I.A.S.: conceptualization, designing of figures and drafting of the original manuscript; M.K.M.: nano-based drug delivery system and life circle of COVID-19; A.V.: approved the final version of the manuscript; A.D.: nanovaccines in clinical trial racing against COVID-19. All authors have read and agreed to the published version of the manuscript.

**Funding:** This research received no external funding.

**Acknowledgments:** We acknowledge the contributions of Ajeet Kaushaki, Gunjan Singh, Nura Mustapha, Yusuf Baba Dala, Salisu Abdullahi Sani, Murtala Muhammad, Saudat Yau Ahmad, and Mashahud Shaarani Yusuf.

**Conflicts of Interest:** The authors declare no conflict of interest.

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
