# Peer review of "Application of Nanotechnology in COVID-19 Infection: Findings and Limitations"

_jnt, doi:10.3390/jnt3040014_

Round 1

Reviewer 1 Report

In this manuscript, the authors aimed to summarize the application of nanoparticles and nanomaterials for the development of diagnostic devices, anti-viral drugs, vaccines, and personal protective equipment to combat COVID-19. This is a hot topic and very interesting. However, the current manuscript suffers from many limitations as stated below, and needs to be re-written for publication in any scientific journals.

Many errors exist in grammar, spelling, sentence structure, and logic throughout the manuscript and the manuscript needs careful editing by someone with expertise in technical English. A lot of statements/claims in this manuscript are overstated, not scientifically solid, and referenced by inappropriate/wrong references or lack of references. Many previous studies published before the Year 2019 were used as COVID-19 examples. Some conclusions are overstated. Discussion on the Covid-19 vaccine status and Table 3 need to be updated.

Despite a large quantitation of information presented, the manuscript seems not scientifically solid and has lots of unclear redundant/overlapping statements and sentences without references.  Particularly, the structures in Part 4/5/6 are logically disordered. Anti-viral nanodrugs and nano vaccines function in different mechanisms and should be discussed separately. I suggest it is better to reconstruct Part 4-6, by re-arranging the samples to discuss the specific subtopic.

Considering the manuscript focuses on nanotechnology to combat COVID-19, the authors should summarize and classify these nanoparticles/nanomaterials in terms of different applications. The manuscript would benefit from significant editing and focus more on the rationale, unique advantages, and mechanisms to reflect the conceptual and technical progression of nanotechnology-based strategies.

Author Response

Reviewer one

Thank you for appreciating our contribution, pls find my responses below

Many errors exist in grammar, spelling, sentence structure, and logic throughout the manuscript and the manuscript needs careful editing by someone with expertise in technical English. A lot of statements/claims in this manuscript are overstated, not scientifically solid, and referenced by inappropriate/wrong references or lack of references. Many previous studies published before the Year 2019 were used as COVID-19 examples. Some conclusions are overstated. Discussion on the Covid-19 vaccine status and Table 3 need to be updated.

Response: We revised the manuscript accordingly using track changes and a yellow colour mark

Despite a large quantitation of information presented, the manuscript seems not scientifically solid and has lots of unclear redundant/overlapping statements and sentences without references.  Particularly, the structures in Part 4/5/6 are logically disordered. Anti-viral nanodrugs and nano vaccines function in different mechanisms and should be discussed separately. I suggest it is better to reconstruct Part 4-6, by re-arranging the samples to discuss the specific subtopic.

Response: The discussion in sect 4-6 aimed to highlight the common application of different nanomaterials on covid-19 vaccine transmission prevention through surface sterilization and air filtration. It is not meant to discuss their MOA, we highlighted the mechanism of their action on page 20, table 3 “Nanoparticles and their mechanism of targeting and detecting SARS-CoV-related viruses changes”

Considering the manuscript focuses on nanotechnology to combat COVID-19, the authors should summarize and classify these nanoparticles/nanomaterials in terms of different applications. The manuscript would benefit from significant editing and focus more on the rationale, unique advantages, and mechanisms to reflect the conceptual and technical progression of nanotechnology-based strategies.

Response: We revised the manuscript accordingly using track changes and a yellow colour mark

Reviewer 2 Report

The manuscript by Ibrahim A. Shehu et al. titled "Emergence of Nanotechnology as Gold Standard to Combat COVID-19 Infection" was submitted to "Journal of Nanotheranostics" for publication. The authors presented a review about the latest advances in the use of nanotechnology to fight against the SARS-CoV-2 virus. In the first section, the authors made a detailed description of the different nano-assisted druggable targets followed by the next sections describing the different strategies for sensing and diagnosis, the different ways for the inactivation of the virus, and finishing the review with the nano-based materials for treatment and vaccine production. Overall, the topic and the content of the review fit well within the scope of the journal. However, before its final acceptance for further publication in the Journal, major revisions have to be performed. Below, some recommendations for the manuscript improvement are described:

First, there are some misspellings, incomplete sentences and typographical errors throughout the manuscript. The manuscript has to be reviewed in deep. It is necessary to review all the figures and significantly improve their quality.

Some of the mistakes and recommendations:

a)  In the Abstract

-Lack of brackets for the used acronyms: (POC), (AgNPs).

-Incomplete sentences: “In addition to the emergence of anti-viral materials for personal protection, nano-assisted vaccines for effective management of COVID-19 infection

-The authors refer to the “second wave of infection”. This is not updated as most countries worldwide are experiencing at least 4 waves. As this is something highly dependent on each country, I will suggest avoiding the mention of a number of infection waves and better using the number of infections (people infected) worldwide or for big regions.

b)      Section 1:

-The introduction must be updated, especially regarding the data offered: “The incumbent cases in the fourth week of July 2021 indicated the COVID-19 infection affected over 190 million people, with a mortality of about 4.1 million across the global population.”

-Please use % instead of “percent” or “per cent”. This should be checked in the whole manuscript.

-Lack of brackets for the used acronyms, some of them: (LNPs).

-Some acronyms must be described as they are the first time appearing in the main manuscript even though they previously appeared in the abstract): AI, AgNPs… This should be checked in the whole manuscript.

c)       Section 2:

-Incomplete sentences: Page 2, first paragraphRNA transcription and replication mediated by helicase and RdRp. The means of transla- 65 tion and proteolytic of viral proteins mediate by chymotrypsin-like and papain-like pro- 66 teases. In addition to virion assembling and exocytosis.”

-I strongly recommend summarizing all the main targets described in a table, indicating the type, brief description and role.

d)      Section 3:

-Incomplete sentences: Page 4, second paragraphThe presence of COVID – 19 is using nanopores of biosensors. [41]

-In some cases, the authors used SARS-COV-2 instead of SARS-CoV-2.

-Missing reference: Page 5, Section 3.1.1 “Lv et al. and Ju et al. 201 examined 15 and samples collected from COVID-19 patients, respectively, and found a high amount of specific antibodies against SARS-CoV-2 RBD.”

e)       Section 6:

-The following data must be updated as it is referred to a Situation Report dated in 2020:

More than 158 vaccine candidates are in the clinical trial racing against COVID-19, and over 135 candidates have entered stage III. [222] For instance, the University of Cambridge /PharmaJet and Wuhan Institute of Biological Products and Sinopharm’s vaccines have been approved for emergency use in China and the United Arab Emirates UAE. Likewise, the CoronaVac (formerly PiCoVacc) vaccine developed by Sinovac in collaboration with Instituto Butantan and BioPharma received FDA emergency approval for use in China [223] among others.

 -Typo: Page 21 last paragraph “using virus-like nanoparticles for the Development of high immunogen- 743 icity SARS-CoV-2 vaccines

Author Response

Thank you for commenting and contributing to our input. Please find my responses below

Reviewer TWO

In the Abstract

-Lack of brackets for the used acronyms: (POC), (AgNPs).

Response: We revised the manuscript accordingly using track changes and a yellow colour mark

-Incomplete sentences: “In addition to the emergence of anti-viral materials for personal protection, nano-assisted vaccines for effective management of COVID-19 infection”

Response: We have restructure the sentance accordingly using track changes and a yellow colour mark

-The authors refer to the “second wave of infection”. This is not updated as most countries worldwide are experiencing at least 4 waves. As this is something highly dependent on each country, I will suggest avoiding the mention of a number of infection waves and better using the number of infections (people infected) worldwide or for big regions.

 Response: We revised the manuscript accordingly using track changes and a yellow colour mark

  1. b)      Section 1:

-The introduction must be updated, especially regarding the data offered: “The incumbent cases in the fourth week of July 2021 indicated the COVID-19 infection affected over 190 million people, with a mortality of about 4.1 million across the global population.”

-Please use % instead of “percent” or “per cent”. This should be checked in the whole manuscript.

-Lack of brackets for the used acronyms, some of them: (LNPs).

-Some acronyms must be described as they are the first time appearing in the main manuscript even though they previously appeared in the abstract): AI, AgNPs… This should be checked in the whole manuscript.

  Response: We revised the manuscript accordingly using track changes and a yellow colour mark

  1. c)       Section 2:

-Incomplete sentences: Page 2, first paragraph “RNA transcription and replication mediated by helicase and RdRp. The means of transla- 65 tion and proteolytic of viral proteins mediate by chymotrypsin-like and papain-like pro- 66 teases. In addition to virion assembling and exocytosis.”

-I strongly recommend summarizing all the main targets described in a table, indicating the type, brief description and role.

  Response: We summarizing all the main targets in  table 1, and a yellow colour mark

  1. d)      Section 3:

-Incomplete sentences: Page 4, second paragraph “The presence of COVID – 19 is using nanopores of biosensors. [41]”

-In some cases, the authors used SARS-COV-2 instead of SARS-CoV-2.

-Missing reference: Page 5, Section 3.1.1 “Lv et al. and Ju et al. 201 examined 15 and samples collected from COVID-19 patients, respectively, and found a high amount of specific antibodies against SARS-CoV-2 RBD.”

 Response: We revised the manuscript  and updated the missing references accordingly using track changes and a yellow colour mark

  1. e)       Section 6:

-The following data must be updated as it is referred to a Situation Report dated in 2020:

“More than 158 vaccine candidates are in the clinical trial racing against COVID-19, and over 135 candidates have entered stage III. [222] For instance, the University of Cambridge /PharmaJet and Wuhan Institute of Biological Products and Sinopharm’s vaccines have been approved for emergency use in China and the United Arab Emirates UAE. Likewise, the CoronaVac (formerly PiCoVacc) vaccine developed by Sinovac in collaboration with Instituto Butantan and BioPharma received FDA emergency approval for use in China [223] among others.”

 -Typo: Page 21 last paragraph “using virus-like nanoparticles for the Development of high immunogenicity SARS-CoV-2 vaccines”

Response: We revised the manuscript accordingly using track changes and a yellow colour mark

Round 2

Reviewer 1 Report

In my opinion, the authors did not carefully revise this manuscript. I suggest a resubmission of this manuscript after a major revision.

1. Many errors exist in grammar, spelling, and sentence structures throughout the manuscript. The manuscript needs careful editing. For example, in the second paragraph on Page 6,

“his team developed Au-NP -an based biosensor” should be “his team developed an AuNP-based biosensor”. “while inducing interferons production., Therefore, Zn-NPs could be beneficial in the combat against COVID-19 [52,53]”, the extra “,” after “production” should be deleted, and “.” should be added after “COVID-19”. “(qRT-PCR testing). Because; Nucleic acid-based sensing is more handy and economical than qRT-PCR For instance,” should be changed to “(qRT-PCR testing), because  nucleic acid-based sensing is more handy and economical than qRT-PCR. For instance,”

Other obvious errors include but are not limited to,

On Page 5, “Table one: Possible nano-assisted drugable targets of COVID-19” should be changed to “Table 1. Possible nano-assisted druggable targets of COVID-19.” In addition, the table legend should be placed in front of the table.

On Page 15, in Figure 4 legend, “CuONP is a Copper oxide nanoparticle, ZnONP is a Zinc oxide nanoparticle, cBA-NP is a Boric acid nanoparticle, GONP is a Graphene oxide nanoparticle, COD-NP is a Carbon oxide nanoparticle, AuNP is a Gold nanoparticle, AgNP is Sliver nanoparticle, and QDs is Quantum dots” should be revised to “CuONP is a copper oxide nanoparticle, ZnONP is a zinc oxide nanoparticle, cBA-NP is a boric acid nanoparticle, GONP is a graphene oxide nanoparticle, COD-NP is a carbon oxide nanoparticle, AuNP is a gold nanoparticle, AgNP is sliver nanoparticle, and QDs are quantum dots”.

On page 15 Lines 433-435, “nanoparticles of different metals and metal oxides are extremely promising to kill viruses through different mechanisms [155]. As schematically presented in figure 4.” should be revised to “nanoparticles of different metals and metal oxides are extremely promising to kill viruses through different mechanisms [155], as schematically presented in Figure 4.” Besides, the GONPs in Figure 4 are not either metals and metal oxides, as claimed by the authors.

On Lines 451-452, “interferon-gamma (IFN-δ) and necrotic tumour factor. (TNF-α).” should be revised to “interferon-gamma (IFN-γ) and necrotic tumour factor-alpha (TNF-α).”

2. Lots of inappropriate/wrong references were cited throughout this manuscript.

For example, on Page 5, “The detection can easily be achieved since the nano biosensors coupled with an isothermal amplification assay can confirm the presence of COVID - 19 in less than 40 minutes. [39,40]”, the authors used articles published before 2014 to indicate the application of nano biosensors to detect COVID-19 virus?

“For instance, CNT-based biosensors were developed to detect the nucleocapsid protein of COVID - 19 with high sensitivity in the sample.” “However, photonic crystals functionalized with amino and carboxyl functional groups can detect SARS-COV-2 spike” Appropriate references are lacking for these examples.

On page 16, “Staroverov et al. reported that encapsulation of COVID-19 protein fragment in AuNP might be considered a potential antiviral agent for vaccine development, [158,159] [160]”, the reference articles were published before the year of 1988, how could they use the COVID-19 protein fragment?

3. The logic and classification of this manuscript should be improved to make the manuscript scientifically solid. Particularly Part 4/5/6 are logically disordered and need to be rearranged. In the current version, examples of anti-viral nanodrugs, and nano vaccines are all mixed-up and discussed in every part repeatedly in Part 4/5/6. The unclear and overlapping statements and sentences made this manuscript not readable. The authors should revise these parts to make the logic clear. It is kind of overstated to claim “nanotechnology as a gold standard to combat Covid-19 infection”.

Author Response

Dear reviewer, thank you for your contributions and comments. We have made substantial revision to the manuscript according to your comments. We therefore, restructured the section 4/5/6 using track changes. 

As attached to this

submission.

Reviewer 2 Report

The major issues were properly addressed.

Author Response

Dear reviewer, thank you for recognizing our effort, thanks for your time and consideration.